# Trading robust representations for sample complexity through self-supervised visual experience

Andrea Tacchetti*          Stephen Voinea          Georgios Evangelopoulos†

The Center for Brains, Minds and Machines, MIT
McGovern Institute for Brain Research at MIT
Cambridge, MA, USA
{atacchet, voinea, gevang}@mit.edu

## Abstract

Learning in small sample regimes is among the most remarkable features of the human perceptual system. This ability is related to robustness to transformations, which is acquired through visual experience in the form of weak- or self-supervision during development. We explore the idea of allowing artificial systems to learn representations of visual stimuli through weak supervision prior to downstream supervised tasks. We introduce a novel loss function for representation learning using unlabeled image sets and video sequences, and experimentally demonstrate that these representations support one-shot learning and reduce the sample complexity of multiple recognition tasks. We establish the existence of a trade-off between the sizes of weakly supervised, automatically obtained from video sequences, and fully supervised data sets. Our results suggest that equivalence sets other than class labels, which are abundant in unlabeled visual experience, can be used for self-supervised learning of semantically relevant image embeddings.

## 1   Introduction

Transformation invariance and learning in small sample regimes are among the most remarkable abilities of the human perceptual system, and arguably the ones that have proven most difficult to replicate in artificial systems. For example, while humans effortlessly recognize new faces after seeing a single image, despite changes in pose, illumination, or facial expression, convolutional neural networks require thousands of examples to achieve similar degrees of generalization. Crucially, these two abilities are mathematically and computationally related in the sense that robust representations of perceptual input support low-sample generalization [1, 6, 27, 8, 30].

Neuroscientists have long debated whether exposure to specific modes of visual experience, such as spatial proximity or sequential presentation, is necessary to learn visual representations that are robust to complex transformations, or if most of these abilities are innate and independent of visual experience [23, 24, 29]. Experiments on chicks reared in highly controlled visual environments revealed that being exposed to temporally smooth object transitions during development is necessary to acquire robustness to 3D rotations [36]. Similarly, newborn monkeys deprived of any visual experience of faces were found not to exhibit a face selective cortical area [4]. These results highlight that exposure to a naturalistic visual experience is key to learning invariant representations, and more in general to the development of a powerful visual system.

Motivated by these findings, we aim to bridge the sample complexity divide between artificial and biological perception systems. We argue that while humans can learn new visual concepts from few examples, they do so by relying on the rich and exhaustive visual experience acquired during and

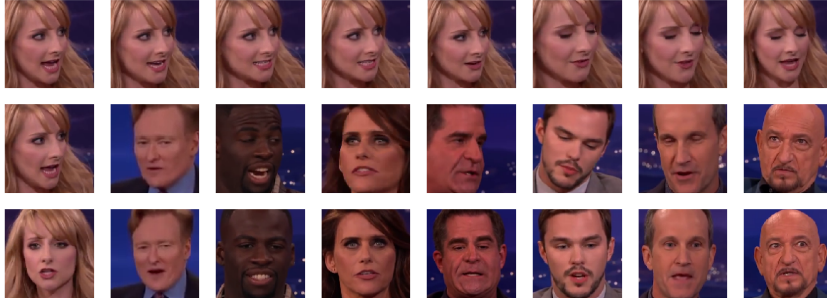

Figure 1: **Image orbits for self-supervised learning from videos**: Images of an orbit sequence from the "Late Night" video face dataset (top row), random samples from distinct orbits (middle row) and their detected canonical, frontal view (bottom row).

after development. By contrast, artificial systems typically rely on explicit supervision of individual instances for learning representations and prediction functions from their inputs. In this work, we consider the idea of allowing artificial systems to learn representations of images through weak supervision, prior to performing supervised visual tasks in low sample regimes. We consider as a natural source of weak supervision the existence of classes or groupings in the input space that are not necessarily related to any downstream learning task, but which encode equivalence relations. Examples are the set of images of an object under rotations [2, 14] or the frames of a video of a moving object [20, 40, 33]. More in general, these equivalence relations can be, temporal, categorical or generative, and partition the space in sets which we will loosely refer to as *orbits* in this paper.

In order to explicitly use the information in such partitions of a training set, we introduce a new loss function that promotes representations that are robust to inter-orbit and selective to intra-orbit relations. Using the same, deep convolutional parametrization, we compare and contrast our approach to reference methods: the supervised triplet loss [25], the ranking loss [33], the surrogate class loss [10] and, when a full model of the orbit-generating process is available (e.g. affine transformations), spatial transformer networks [16].

We demonstrate how image embeddings learned through weak supervision using our novel loss function and orbit sets support one-shot learning in a variety of settings and reduce the sample complexity of challenging recognition tasks. Furthermore, we establish the existence of a trade-off between the sizes of weakly supervised and fully supervised data sets. Overall, our results suggest that partitioning observations into equivalence sets, similar to what is achieved through unlabeled visual experience or self-supervision, and using the proposed loss, one can learn semantically relevant embeddings and lower the sample complexity of downstream visual tasks.

**Related work**: Theoretical aspects of representations invariant to transformations have been studied through the properties of parameterization choices such as convolutions [1, 6, 27, 8, 3]. Robustness and equivariance has been also sought through explicit representation [38, 15] or estimation of transformation parameters.

Similarity information has been used as weak supervision in variants of the triplet loss function for deep metric [28] and embedding learning [33]. Supervised versions of the triplet loss have been used for discriminatively-trained metric learning, through convolutional neural networks (CNNs), by minimizing the true objective of the task, e.g. face verification [7, 25]. The triplet loss was also used for learning transformation-invariant embeddings [14].

Methods using self-supervision for learning or pre-training representations exploit temporal continuity [20, 40, 12, 33, 17, 34], spatial proximity (context, inpainting) [9] or data manipulation (transformation, colorization) [18]. The latter include predicting rotations applied to images [11] and matching pixels between sheared and original images [22]. Defining surrogate classes populated by data augmentation transformations was explored in the exemplar CNN framework [10].

## 2 Definitions and related loss functions

Let $\Phi : \mathcal{X} \to \mathcal{F}$ be a representation on input space $\mathcal{X}$, selected from some hypothesis space $\mathcal{H} \subseteq \{\Phi \,|\, \Phi : \mathcal{X} \to \mathcal{F}\}$. We will assume $\mathcal{X}$ and $\mathcal{F}$ to be Euclidean spaces, and that elements of $\mathcal{H}$ can be parameterized for learning (e.g. $\Phi$ is a deep CNN). Embedding maps $\Phi$ can be learned by

minimizing a loss function $L$ that is suitable for the task at hand. Loss functions can be defined on the input space $L : \mathcal{X} \times \mathcal{X} \to \mathbb{R}_+$, the feature space $L : \mathcal{F} \times \mathcal{F} \to \mathbb{R}_+$, or the output space $L : \mathcal{Y} \times \mathcal{Y} \to \mathbb{R}_+$; moreover loss functions can be combined to provide more structure to the learning problem.

**Transformations and orbit sets** A family of transformations is a set of maps $\mathcal{G} \subset \{g \mid g : \mathcal{X} \to \mathcal{X}\}$ where $gx$ is the action of the transformation. Transformations in $\mathcal{G}$ act on $x$ to generate points $x'$, i.e. $x' = gx = g(x)$. Transformations can be parametrized by $\theta_j \in \Theta$, so that $\mathcal{G} = \{g_j = g_{\theta_j} \mid \theta_j \in \Theta\}$ (e.g. rigid translation or rotation across the center).

**Definition 1** (Group orbits [1]). *An orbit associated to an element $x \in \mathcal{X}$ is the set of points that can be reached under the transformations $\mathcal{G}$, i.e., $\mathcal{O}_x = \{gx \in \mathcal{X} \mid g \in \mathcal{G}\} \subset \mathcal{X}$.*

**Triplet loss**: The triplet loss [25] enforces that $\Phi$ maps points into the embedding space so that pairwise intra-class dissimilarities are smaller than pairwise inter-class dissimilarities. Let $(x_i, x_j)$ be image pairs, then by defining triplets $\mathcal{T} = \{(x_i, x_p, x_q) \mid y_i = y_p, y_i \neq y_q\}$, where $y_i \in \mathcal{Y}$ is the class label, the loss enforces

$$L(\Phi(x_i), \Phi(x_p)) \leq L(\Phi(x_i), \Phi(x_q)) - \alpha \tag{1}$$

by minimizing a variant of the large margin loss [35]:

$$\min_{\Phi \in \mathcal{H}} \sum_{i=1}^{|\mathcal{T}|} |L(\Phi(x_i), \Phi(x_p)) + \alpha - L(\Phi(x_i), \Phi(x_q))|_+ , \tag{2}$$

where $\alpha \in \mathbb{R}_+$ a distance margin for the non-matching pairs, $|\cdot|_+ = \max\{0, \cdot\}$ is the hinge loss and $L$ is a measure of dissimilarity in the output space (e.g. Euclidian distance).

**Reconstruction loss**: In addition to the encoding map $\Phi : \mathcal{X} \to \mathcal{F}$ in $\mathcal{H}$, we can consider a corresponding decoding map $\tilde{\Phi} : \mathcal{F} \to \mathcal{X}$ in $\tilde{\mathcal{H}}$ and minimize the reconstruction or autoencoder error:

$$\min_{\Phi \in \mathcal{H}, \tilde{\Phi} \in \tilde{\mathcal{H}}} \sum_{i=1}^{n} L(x_i, \tilde{\Phi} \circ \Phi(x_i)) \tag{3}$$

over $n$ training points. The two maps usually have related structures (e.g. tied weights, convolutional-deconvolutional) [5]. The loss function $L$ is again a measure of dissimilarity in the input space (e.g. Euclidean distance or mean squared error).

**Exemplar loss**: A surrogate class can be formed by applying random transformations sampled from some known $\mathcal{G}$ to each point in an unlabeled set $\{x_i\}_{i=1}^n$. The exemplar loss [10] minimizes a discriminative loss $L$ (e.g. the cross-entropy loss) with respect to these surrogate classes:

$$\min_{\Phi \in \mathcal{H}, f \in \mathcal{H}_f} \sum_{i=1}^{n} \sum_{j=1}^{k_i \leq |G|} L(i, f(\Phi(g_j x_i))) \tag{4}$$

where $i$ indexes the original, untransformed training set and serves as the surrogate class label for points generated from $x_i$; $f$ is a classifier learned jointly with the embedding $\Phi$. By learning to classify transformed images according to their source untransformed example, the exemplar loss promotes the learning of embeddings that are robust to transformations in $G$.

**Spatial transformer networks (STNs)** When a plausible forward model of the action of $g \in \mathcal{G}$ is known and has a suitable parametrization, STNs [16] learn to undo transformations in $\mathcal{G}$.

## 3 Representation learning using a novel loss for transformation sets

Let $\mathcal{X}_n = \{x_i\}_{i=1}^n \in \mathcal{X}$ be a set of unlabeled instances and assume $\mathcal{X} = \mathbb{R}^d$, for example $x_i$ being vectorized images. We aim to learn an embedding $\Phi : \mathcal{X} \to \mathcal{F}$, in some $\mathcal{F} = \mathbb{R}^k$ with $k \leq d$, such that the corresponding metric

$$D(x, x') = ||\Phi(x) - \Phi(x')||_{\mathcal{F}}^2, \quad D : \mathcal{X} \times \mathcal{X} \to \mathbb{R}_+ \tag{5}$$

where $|| \cdot ||_{\mathcal{F}}$ denotes the norm in $\mathcal{F}$, is robust to transformation sets and selective for sets that are not equivalent up to transformations. The above condition is equivalently written as:

$$x' \sim x \Leftrightarrow x, x' \in \mathcal{O}_x \Leftrightarrow D(x, x') \leq \epsilon, \tag{6}$$

where we use an $\epsilon$-approximation to the zero-norm distance for exact invariance, and $\mathcal{O}_x$ is a generic orbit-set (see below). Note that the requirement for selectivity, i.e., the converse direction, makes $(\mathbb{R}^d, D)$ a proper metric space.

### 3.1 Orbit sets for weak- or self-supervision

Following Definition 1, orbits are sets stemming from equivalence relations in $\mathcal{X}$. For example, given a group structure on the transformations $\mathcal{G}$, the input space $\mathcal{X}$ is partitioned into orbits as $x \sim x' \Leftrightarrow \exists g \in \mathcal{G} : x' = gx, \forall x, x' \in \mathcal{X}$. We relax this definition to include set memberships provided by categorical labels as well as other transformations that do not enjoy a group structure.

**Definition 2** (Generic orbits). *An orbit associated to $x \in \mathcal{X}$ is the subset of $\mathcal{X}$ that includes $x$ along with an equivalence relation: $\mathcal{O}_x = \{x' \in \mathcal{X} | x \sim x'\} \subset \mathcal{X}$, given by a function $c : \mathcal{X} \to \mathcal{C}$ such that $x \sim x' \Leftrightarrow c(x) = c(x')$.*

Examples of $c$ are the labels of a supervised learning task, the indexes of vector quantization codewords or, for the case of sequential data such as videos, the (sub)sequence membership, with $\mathcal{C}$ the set of classes, codewords or sequences respectively (as in Fig. 1, row 1). Generic orbits can be obtained from data in an explicit, e.g. by transforming points, or an implicit way, e.g. through auxiliary tasks and groupings using weak-supervision. Here are some examples:

**Virtual examples** Given a parametrized set of transformations, orbits can be generated by randomly sampling from the parameter vectors $\{\theta_j \in \Theta\}$ and letting $\mathcal{O}_x = \{g_{\theta_j} x \,|\, \theta_j \in \Theta\}$ for a given $x$. Examples include geometric transformations, e.g. rotation, translation, scaling [21] or typical data augmentation transforms, such as cropping, contrast, color, blur or illumination [10].
**Acquisition** If the data acquisition process can be designed *ad hoc*, or if its characteristics are encoded in meta-data, e.g. multiple samples of an object across time, conditions or views, then an orbit can be associated to all samples from a session [13].
**Self-supervision** For sequential data such as videos, an orbit can be a continuous segment of the video stream, or an object/object part detected and tracked across time [33], following plausible expectations on feature smoothness and continuity of the representation [40, 17].

In the following, for a given $\mathcal{X}_n$, we obtain the set of orbits $\{\mathcal{O}_{x_i}\}$ either via an auxiliary supervision signal if available (video sequence, or images session index) so that $\mathcal{X}_n = \cup_{x_i} \mathcal{O}_{x_i}$, or by augmentation of each $x_i \in \mathcal{X}_n$ so that $\mathcal{O}_{x_i} = \{g_{\theta_j} x \,|\, \theta_j \in \Theta\}$. We further assign a *canonical example* $x_c \in \mathcal{O}_{x_i}$ to each orbit.

**Definition 3** (Orbit canonical example). *The canonical example is an arbitrary chosen point in the orbit set that provides a reference coordinate system for the transformations and is consistent across orbits.* For $\mathcal{O}_x$ obtained through virtual examples, $x_c$ is the result of the identity transformation $g_0 \in \mathcal{G}$, i.e. $x_c = g_0 x = x$. Otherwise, $x_c$ is chosen or detected as the no-pose or neutral condition example (Fig. 1, bottom row).

### 3.2 Loss function

Given the training set orbits, we define a set of triplets of points

$$\mathcal{T} \subset \{(x_i, x_p, x_q) \,|\, x_i \in \mathcal{X}_n, x_p \in \mathcal{O}_{x_i}, x_q \in \mathcal{O}_{x_q}; \mathcal{O}_{x_i} \cap \mathcal{O}_{x_p} = \emptyset\} \quad (7)$$

such that each $x_i$ is assigned a *positive example* $x_p$ (in-orbit), i.e. $x_i \sim x_p \Leftrightarrow x_i, x_p \in \mathcal{O}_{x_i}$ and a *negative example* $x_q$ (out-of-orbit), i.e. $\mathcal{O}_{x_q} \cap \mathcal{O}_{x_i} = \emptyset$. The proposed loss function is composed of two terms; (1) a discriminative term, based on the triplet loss, using distances between the encodings $\Phi : \mathbb{R}^d \to \mathbb{R}^k$ on the feature space $\mathbb{R}^k$

$$L_t(x_i, x_p, x_q) = \left| \|\Phi(x_i) - \Phi(x_p)\|_{\mathbb{R}^k}^2 + \alpha - \|\Phi(x_i) - \Phi(x_q)\|_{\mathbb{R}^k}^2 \right|_+, \quad (8)$$

with $\alpha$ a distance margin, and (2) a reconstruction error between the decoder output $\tilde{\Phi} : \mathbb{R}^k \to \mathbb{R}^d$ and the orbit canonical, as a distance on the input space $\mathbb{R}^d$

$$L_e(x_i, x_c) = \left\| x_c - \tilde{\Phi} \circ \Phi(x_i) \right\|_{\mathbb{R}^d}^2, x_c \in \mathcal{O}_{x_i}. \quad (9)$$

The representation learning problem is then formulated as

$$\min_{\Phi, \tilde{\Phi}} \sum_{i=1}^{|\mathcal{T}|} \left( \frac{\lambda_1}{k} L_t(x_i, x_p, x_q) + \frac{\lambda_2}{d} L_e(x_i, x_c) \right), \quad (10)$$

where the hyperparameters $\lambda_1, \lambda_2$ control the relative contribution of the two terms.

|  | **Embedding** | **Train/Test/Validation** | |
|---|---|---|---|
| MNIST | Train x 32 | Test x 32: 10-fold | |
| Even/Odd MNIST | Even Train x 32 | Odd Test x 32: 10-fold | |
| NORB | Train (25 objects) | Test (15/10 objects, 1000 re-splits) | |
| Multi-PIE | 2 Sessions (4 choose 2) | 2 Sessions (4 choose 2) | |
| Late Night | 500 sequences | 28 x 3 sequences | |

|  | **Transformations** | **Orbits** | **Canonical** |
|---|---|---|---|
| MNIST | 2D affine | Random affine (32) | Original |
| Even/Odd MNIST | 2D affine | Random affine (32) | Original |
| NORB | 3D pose | 3D pose (162) | Frontal |
| Multi-PIE | 3D viewpoint | 3D viewpoint (13) | Frontal |
| Late Night | Unconstrained | 500 detected sequences | Min yaw face |

Table 1: Datasets, orbits and canonical definitions used in our experimental evaluations.

### 3.2.1 Orbit triplet (OT) loss

For $\lambda_2 = 0$ the loss reduces to the triplet loss in Eq. (2), with the orbit identity as label. Points that lie on the same orbit are pulled together and points on different orbits are pushed apart. The minimizer is driven to Eq. (1), using all triplets in the training set, which is satisfied in the theoretical minimum of $L_t$. The orbit triplet loss resembles a *Siamese network* architecture [7], with a tied weights embedding trained using triplets as input.

### 3.2.2 Orbit encoder (OE) rectification loss

For $\lambda_1 = 0$ the loss reduces to the reconstruction error of the canonical $x_c$ of the orbit $\mathcal{O}_{x_i}$ by the decoded $\Phi(x_i)$. Equivalently, this is the error of the rectification that $\tilde{\Phi} \circ \Phi : \mathcal{X} \to \mathcal{X}$ applies on the input $x_i$, which is some transformation of $x_c$. This is a novel autoencoder loss, a generalization of denoising autoencoders (which promote reconstruction of clean versions from noisy inputs) to transformations with or without an explicit generative model. In this case the encoder-decoder pair $\tilde{\Phi} \circ \Phi$ learns to de-transform all elements of an orbit by aligning arbitrary samples (Fig. 1, middle row) to the orbit canonical element (Fig. 1, bottom row).

Conceptually, the orbit encoder loss enforces selectivity on $\Phi$ by preserving sufficient information to reconstruct the input irrespective of the transformation. For computing the loss across pairs $(x_i, x_c)$ from different orbits, the choice of $x_c$ has to be consistent only across orbits of the same semantic class of a downstream task, e.g. faces or numbers, and is otherwise arbitrary.

## 4 Experimental evaluation of learned image representations

We systematically evaluate representations learned from different datasets and applied to different visual recognition tasks using the same parametrization, different loss functions, and varying degrees of supervision (unsupervised, supervised, weakly-supervised). Our experimental procedure is explicitly designed to probe the impact of learning robust representations using generic orbit sets on the performance in downstream tasks. To this end, we separate the representation learning step from the supervised learning task and purposefully choose extremely low samples (e.g. 1 example per class) and simple classifiers (e.g. 1-NN) for the latter. Moreover, we select datasets where defining orbit sets is intuitive and done by either generating virtual examples, or by exploiting the characteristics of the data acquisition process. Table 1 provides an overview of the datasets, transformations, orbit and canonical examples definitions in our evaluations. Overall, the sets include 3D viewpoint, light and unconstrained transformations; MNIST is the only one with analytic, known transformation orbits.

**Evaluation procedure**: For each dataset, we employ an *Embedding set* to learn representations and a separate *Validation set* for determining the optimal number of SGD iterations (early stopping hyperparameter) by maximizing the mean, across multiple re-splits of the set, of task-specific performance metrics. The representations are used for encoding *Train and Test sets* from a *Supervised set* for the downstream recognition task. Performance is reported as the mean/std of multiple train/test re-splits of the *Supervised set*. For all experiments, there is no overlap between embedding, validation, and supervised sets, or between train/test splits, within the validation and the supervised sets. These rules were enforced at the level of orbit subsets, a stronger requirement than excluding single samples.

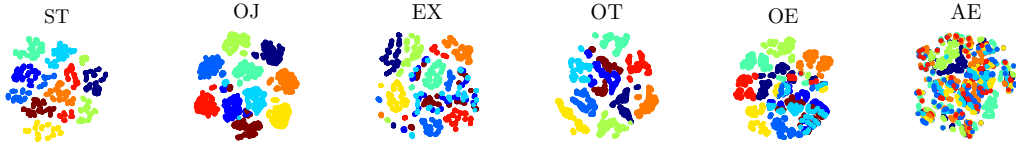

Figure 2: Two-dimensional t-SNE visualizations of face image embeddings from 10 randomly sampled subjects/classes (coded by different colors) of Multi-PIE test set.

**Embeddings**: We compare embeddings learned using the proposed loss, termed Orbit Joint (OJ) from Eq. (10); the two special cases, Orbit Triplet (OT) ($\lambda_2 = 0$) and Orbit Encode (OE) ($\lambda_1 = 0$); and three, closely related losses: standard Autoencoder (AE), Supervised Triplet (ST) [25] and Exemplar (EX) [10]. OT also corresponds, analytically and empirically, to using the ranking loss (RL) [33] with the orbit-based triplets defined in our work. For the affine MNIST evaluations, we also used an embedding featuring a Spatial Transformer Networks module [16], trained with orbit supervision (OT-STN) or full supervision (ST-STN). It is worth highlighting that we included the ST and AE embeddings only as a way of providing a sense for the scale of the quantities we consider. ST is a fully supervised method (i.e. the partitioning of the embedding dataset is defined according to the same class labels as the downstream task) and it provides a performance ceiling to weakly supervised methods (similarly ST-STN). AE on the other hand is purely unsupervised (i.e. there is no partitioning at all, each image stands on its own), and serves as a performance floor.

**Network and training**: For the encoder, we used a deep convolutional network with architecture following VGG networks [26]. Each layer is composed of a series of convolutions with a small $3 \times 3$ kernel (of stride 1, padding 1), batch normalization and Rectifier Linear Unit (ReLU) activations. A spatial max pooling layer (of stride 2 and size either $2 \times 2$ or $4 \times 4$) was used every two such layers of convolutions. The number of channels doubled after each max pooling layer, ranging from 16 to 128 for MNIST, 64 to 512 for Multi-PIE and the Late Night Dataset and 16 to 256 for NORB (image sizes were single channel, $40 \times 40$ px for MNIST, 3-channel, $128 \times 128$ px for Multi-PIE, single channel $96 \times 96$ px for NORB and 3-channel $96 \times 96$ px for the Late Night dataset). Four iterations of convolution and pooling were followed by a final fully-connected layer of size 1024. The decoder is a deconvolutional network, reversing the series of encoder operations, using convolutional reconstruction and max unpooling [37], in direct correspondence to the encoder in the number of layers, filters per layer and size of kernels. Encoder and decoder weights were tied with free biases.

Loss minimization was carried out with mini-batch Stochastic Gradient Descent using the Adam optimizer. For MNIST, we used mini-batches of size 256, for Multi-PIE, 72, for NORB, 256, for Multi-PIE, 72 and for the Late Night dataset, 128 images. The selection of triplets for ST, OT and OJ followed the soft negative selection process from [25]. The values for $\lambda_1$ and $\lambda_2$ were set equal ($\lambda_1 = \lambda_2 = 1$). The STN modules consisted of two max pooling-convolution-ReLU blocks with 20 filters of size $5 \times 5$ (stride 1), pooling regions of size $2 \times 2$ and no overlap, followed by two linear layers. Where applied, these were inserted between the input and the rest of the network.

**Summary of results** A summary of the test set performance in all tasks and sets is provided in Table 2. Notably the performance of OJ is either better or statistically indistinguishable (with a standard significance threshold at $p < 0.05$) from OT and OE. This observation makes the case for the joint loss, which can result in substantial improvements like in the one-shot classification task on MNIST with affine transformations. This also suggests that more optimal, e.g. by cross-validation, selection of the hyperparameters $\lambda_1$ and $\lambda_2$ could lead to further performance gains.

## 4.1 Affine transformations: MNIST

We first validated our method on MNIST using affine transformations to generate orbit sets. This allows us to illustrate our method on a simple dataset with easily defined orbits. We transformed each image in the original set with 32 random affine transformations. Transformations were sampled uniformly from rotations in $[-90°, 90°]$, shearing factor in $[-0.3, 0.3]$, scale in $[0.7, 1.3]$, and translation in $[-15, 15]$ pixels in each dimension. The embedding set consisted of the original training set ($50 \times 10^3$ images), augmented by 32 transformations for each sample, resulting in a total of $1650 \times 10^3$ images. Images were grouped in $50 \times 10^3$ orbits, each one a 33-sized set of the

| | ST [25] | OJ | EX [10] | OT, RL [33] | OE | ST-STN | OT-STN |
|---|---|---|---|---|---|---|---|
| MNIST | 0.97±0.02 | 0.67±0.05 | **0.44±0.06** | **0.37±0.04** | **0.40±0.05** | 0.97±0.01 | 0.34±0.03 |
| Even/Odd | 0.49±0.08 | 0.59±0.08 | **0.55±0.07** | **0.52±0.07** | **0.54±0.07** | 0.51±0.02 | 0.54±0.02 |
| NORB | 0.67±0.13 | 0.59±0.12 | 0.58±0.11 | **0.55±0.12** | 0.54±0.11 | | |
| M-PIE AUC | 0.99±0.00 | 0.95±0.01 | **0.89±0.01** | 0.92±0.01 | **0.87±0.02** | | |
| Precision | 0.99±0.00 | 0.91±0.01 | **0.71±0.04** | 0.93±0.02 | **0.83±0.02** | | |

Table 2: Performance evaluation using embedding-validation-test splits. Entries in **bold** denote significant performance difference between the proposed loss (OJ) and the corresponding weakly supervised losses EX, OT, OE ($p$-values less than 0.05, Bonferroni-corrected paired $t$-test). STN columns denote the use of additional spatial transformer modules [16].

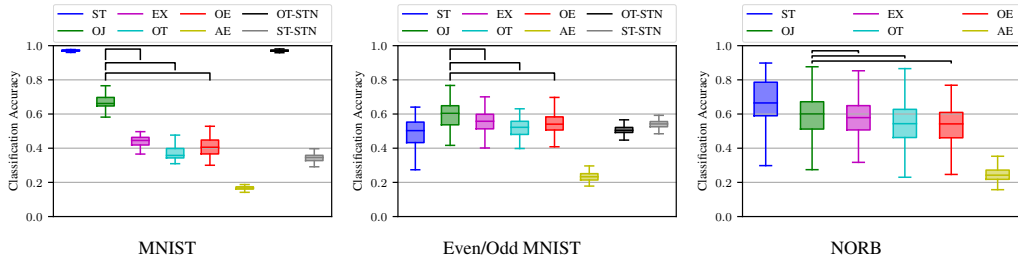

Figure 3: Classification accuracy on one-shot learning (1-NN) using learned image embeddings. Box-plots obtained with multiple train/test splits (Table 2).

original (canonical example) and its random transformations. We also used 10 random re-splits of the augmented test set for the validation and train/test-sets.

The learned embeddings were employed in a one-shot classification task, using one image per class and Nearest Neighbor classification (1-NN). Classification accuracy over the 10 re-splits is shown in Fig. 3 (left). The proposed loss OJ achieved top accuracy among the weakly-supervised methods, followed by EX and OE. Note how the addition of spatial transformer modules provided no improvement either with full or with orbit supervision.

## 4.2 Transfer learning: even/odd MNIST

We designed a simple transfer learning setting by using exclusively images of *even* digits in the embedding set and *odd* digits in all other sets. The task consisted of a new 5-way, one-shot classification and the effect in accuracy can be seen in Fig. 3 (middle). In this case, the OJ loss significantly outperforms the supervised ST. Sampling (randomly) from the transformations and using them to define finer equivalence classes was more helpful than having access to class-level information.

## 4.3 3D-affine transformations: NORB

To assess robustness to changes in illumination and 3D pose, we used the NORB-small dataset [19]. The dataset contains images of 50 toys from 5 generic categories acquired from 162 different 3D viewpoints and under 6 lighting condition. The embedding set consisted of 5 objects per category from the original training set, and the validation and train/test sets from 2 and 3 objects per category from the original test respectively. The task was one-shot classification with 1-NN, with 1000 random re-splits in validation and train/test-sets (Fig. 3 (right)). This is the only dataset in our evaluations in which OJ performance falls on the same range as the orbit triplet and the exemplar loss.

## 4.4 Face transformations: Multi-PIE

The Multi-PIE dataset [13] contains images of faces of 129 individuals, captured from 13 distinct viewpoints and under 20 different illumination conditions. Acquisition was carried out across four sessions, resulting in a dataset of $129 \times 13 \times 20 \times 4 = 134'160$ images. We used six splits (4 choose 2) by session (2 sessions for embedding – 1 for validation and 1 for test). For learning the embeddings,

the ST method had access to the face identity for each image, thus considering equivalence classes formed by all images of the same subject (across sessions, viewpoints and illumination conditions). The weakly-supervised methods (OJ, OT, OE, EX) had only access to the set of $129 \times 20 \times 3$ orbits in the embedding set, each corresponding to all 13 viewpoints for a single identity, illumination condition and session.

Figure 2 shows the relative distance landscape, as 2D t-SNE plots [31], for all images from 10 random subjects of the test set, encoded using embeddings learned with different losses. We used two distance-based tasks to evaluate performance: a *same-different* face verification task and a face retrieval task, measuring the Area Under the ROC Curve (AUC) and the mean top-1 precision respectively. The proposed OJ loss achieves the best AUC score among the weakly-supervised methods (Table 2), and similarly for the top-1 precision scores, with the notable exception of OT outperforming OJ.

For the verification, we used all unique pairwise distances in the embedding space. For the retrieval, we selected the closest point to a query (top-1) from a target set. We considered each test image as query, and the rest of the test set as target set (after removing all same-identity images (32 in total, including the query) at the same illumination (regardless of viewpoint) and at the same viewpoint (regardless of illumination) to ensure we were evaluating the preference of identity over appearance.

## 5 Self-supervised embeddings from videos

Video sequences provide an interpretable and rich form of self-supervision through the temporal progression of a scene, an event, or a moving/transforming object. In this section we focus on exploiting this source of weak supervision to lower the sample complexity of a supervised downstream task. That is, by learning robust embeddings from video sequences, we are able to lower the number of labelled examples required to achieve a fixed performance level in a downstream recognition task.

### 5.1 Late Night face dataset

We collected a dataset of human faces video-clips for learning embeddings and testing recognition and transfer learning. The sources were YouTube video clips from the official channel of "CONAN", a late night TV talk show with face transformations such as 3D pose changes and deformations, e.g. facial expressions and dialogue.

**Automatic orbit extraction**: The collection pipeline was designed to extract contiguous clips of individually cropped faces (Fig. 1, top row) and was as follows: (1) Detect scene changes (i.e. camera cuts) and assign each frame to a unique, uninterrupted sequence (2) Detect and crop faces, independently in each frame, using a high-recall face detector [32]. (3) Enhance detection precision by using a high-precision face detector on the cropped faces [39]. (4) Construct orbits using the detection positions in adjacent frames – regions in subsequent frames were assigned to the same orbit if their bounding boxes overlapped, else, a new orbit was started from the latter detection. (5) Retain largest orbit (number of images) from each video. (6) Choose a canonical example for each orbit as the face image with the least yaw displacement from a frontal pose (estimated by the face detector) – in case of multiple candidates, choose the one with the highest detection confidence score [39].

**Embedding set**: From processing 500 unlabeled source videos, we collected $271'415$ images across 500 orbits, which were not manually validated or quality assured in any way. Crucially, individuals appear in more than one orbit.

**Evaluation set**: We collected 84 orbit clips of 28 faces (3 orbits per identity), not appearing in the embedding set. In this case, we manually assured for quality that all images in each orbit belong to the same identity and labelled the dataset according to face identity. This resulted in $47'919$ images, in 28 categories and 84 orbits.

### 5.2 Sample complexity and embedding complexity

Learned embeddings were used to encode the evaluation set for a 28-class discrimination task using a linear SVM. To measure the gains in sample complexity between the different methods, we randomly sampled 20 training sets (containing 1 to 20 images per class –resampled 10 times) and one test set with 64 images per class (images from the same orbit never appeared in both sets). Results are shown in Fig. 4 (left), where one can note that OJ outperforms all baselines and special cases, in the small

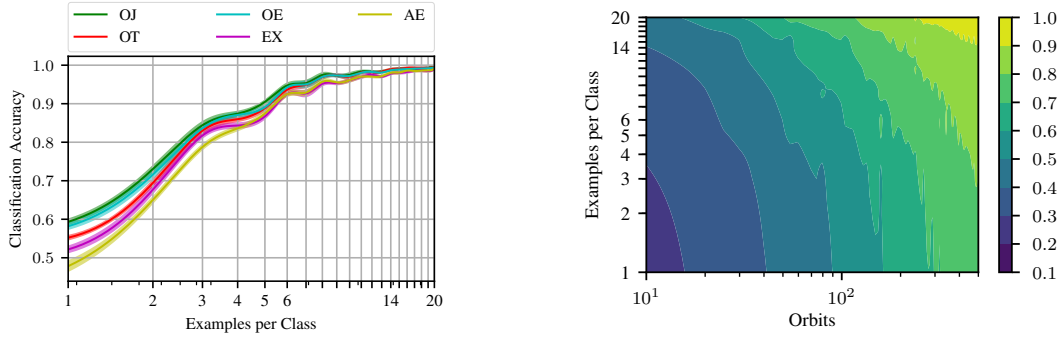

Figure 4: **(left) Sample complexity**: Classification accuracy (mean with standard error across 10 re-samples) vs. training set size on the Late Night face dataset. The task was a 28-way face discrimination (linear SVM), with embeddings learned on a separate set (500 orbits). ST is not available due to the lack of class labeling on the, automatically constructed, embedding set. **(right) Embedding and sample complexity trade-off**: Accuracy map (mean across 10 train/test re-splits of the validation set) of OJ for 1-20 labeled examples per class for classifier learning and 10-500 orbits for embedding learning.

sample regions of the plot, e.g. for classification with 7 samples per class or lower. This comparison also highlights the effect that the representation can have for learning tasks on a label budget.

Next we considered the trade-off in data resources for learning the embedding and learning a predictor on a separate set. In Fig. 4 we demonstrate this as a two-dimensional map of classification accuracy for OJ on the Late Night dataset, by letting the number of labeled examples change between 1-20 per class (as in the sample complexity plot) and the number of orbits between 10-500. Note how the map is dominated by large *equi-accuracy* regimes, which define contiguous performance regions. Moving along those regions, or fixing the performance requirements, is equivalent to trading supervised resources (labelled examples) for the, automatically extracted (change detection, face detection, tracking) and unlabelled, embedding resources. The demonstration of this trade-off is, to the best of our knowledge, a new finding.

## 6 Conclusions

Motivated by findings in the neuroscience of vision on the necessary role of visual experience in the development of robust visual perception, we considered image representation learning using self- or weak- supervision from unlabeled image sets and videos. We proposed a novel loss function that combines a *discriminative* and a *rectification* component of complementary objectives. The scheme supersedes state-of-the-art, exemplar- and ranking-based losses and, when applicable spatial transformer modules, in distance-based and classification tasks. In addition, we find that the learned representations reduce the "label budget" of supervised learning by trading it for "free" self- or weak-supervision. From a practical point, our work suggests that partitioning the training set to equivalence classes defined from sampled or implicitly acquired transformations, is a useful weak-supervision signal for extracting embeddings that are semantically relevant and economically learned. Future work will involve orbit evaluation and sampling on natural and complex image and video datasets.

### Acknowledgments

We would like to thank Tomaso Poggio for his advice and supervision throughout the project and the McGovern Institute for Brain Research at MIT for supporting this research. The DGX-1 used for our experiments was donated by NVIDIA. This material is based upon work supported by the Center for Brains, Minds and Machines (CBMM), funded by NSF STC award CCF-1231216.

## Footnotes

*Currently with DeepMind. †Currently with X, Alphabet.

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
