[Reviews · NeurIPS 2018]

Reviewer 1



This submission describes a model for unsupervised feature learning that is based on the idea that an image and the set of its transformations (described here as an orbit of the image under the action of a transformation group) should have similar representations according to some loss function L. Two losses are considered. One is based on a ranking loss which enforces examples from the same orbit to be closer than those of different orbits. The other one is a reconstruction loss that enforces that an all examples from an orbit should be mapped to a canonical element of the orbit using an autoencoder-like function. Two broad classes of transformation groups are considered, the first one is a set of parametrized image transformations (as proposed by Dosovitskiy et al. 2016) and the other one is based on some prior knowledge from the data - in this case the tracking of faces in a video (as proposed by Wang et al. 2015). The proposed approach is described with a very clear framework, with proper definitions. It makes the paper easy and nice to read but this abundance of definitions and notations may be a bit of a burden for some readers. On a high level, this submission is the generalization of the work of Wang et al. and Dosovitskiy et al.. It gives a more general framework, where different self supervised learning models correspond to different definitions of the orbits. The idea of using two losses, namely the reconstruction and ranking loss makes the framework more general, including the model of Wang et al. and a variant of a denoising autoencoder, where the noise corresponds to samples from the orbit. Even though this paper draws a lot of inspiration from previous work, it fails to compare to them on a common ground. All experiments are ran on MNIST, NORB, Multi-PIE and a video talk show dataset. As baselines, the authors consider [9] and [28] which are the two paper mentioned above. However, none of those methods report numbers on these dataset. I understand that producing experiments on larger scale datasets (as [28]) can be a bit heavy and not accessible to all institutions. However [9] reports an extensive evaluation on STL-10, CIFAR-10 and Caltech-101, which is tractable, even on CPU only machines. Moreover, the setup considered here is 1-shot learning with a NN classifier, resulting in huge confidence intervals and making the results kind of mixed. On Mnist, the there seems to be a strong impact of using both losses (OJ versus OT or OE), justifying the importance of using both of them. I would like to see if this transfers to datasets that were used by [9]. Right now, except on MNIST, the proposed approach falls roughly at the same spot as [9] or [28], making the results hard to interpret. Because the model description is basically just a formal description of previously proposed approaches and that the experimental section does not provide strong evidence for the quality of the model I rate this paper as borderline with a 6. I eagerly await the authors response and the reviewer discussion to make my final decision.

Reviewer 2



This paper proposes a framework for representation learning that doesn't require any labeled data. Instead, priors on invariance to spatial transformations or local invariance in video is used to drive representation learning. This paper offers a lot of well thought out formalism and shows some promise on datasets like one-shot MNIST learning. The main downside is that MNIST is particularly suited for such transformations, and even going to a dataset like NORB, this clear edge over competing methods disappears. To me this is an accept because it gives me a lot ideas (and a fleshd out framework for expressing these ideas) and I think this will be true for the community in general. Strengths. This paper offers a lot of thorough and precise formalism that are general and may find use in a variety of settings. The main idea of defining orbits is interesting. The experiments are thorough and interesting, and the paper is well written. Weaknesses. It's hard to judge impact in real-world settings when most of the quantitative evaluations are on datasets not representative of complex natural images (e.g. MNIST and NORB). On MNIST, the method shows clear advantages over competing methods. However, even on NORB, where a lot of the deformations can't easily be parameterized, this advantage has turned into being only on par with other leading methods. I think the inclusion of the faces dataset was important for this reason. I was confused for a while what the exact orbit was for each dataset. I kept scanning the text for this. A table of all three datasets and a short note on how orbits were defined and canonical samples selected would make things a lot clearer. Concurrent work. Similar ideas of representation learning through transformation priors have appeared in recent work. I don't think it takes away any novelty from this submission, since judging from the dates these is concurrent works. I just thought I would bring your attention to it: - https://openreview.net/pdf?id=S1v4N2l0- (ICLR 2018) - https://arxiv.org/pdf/1804.01552.pdf (CVPR 2018) Minor comments. - eq. 6: what connects the orbit with this loss? I don't see the connection just yet - eq. 7: "x_q not in Oxq!=Oxi" What is this notation "set1 != set2" that seems to imply it forms another set (and not a true/false value) line 136: Oxq \not= Oxi, again, I'm not sure about this notation. I understand what it means, but it looks odd to me. I have never seen this as part of set notation before. - eq. 8: where is x_p, x_q, x_c coming from? Shouldn't the summand be $(x_i, x_p, x_q) \in \mathcal{T}$? The canonical sample x_c is still unclear where it comes from. If x_c is the canonical instance for each orbit, then it also changes in the summation. This is not clear from the notation. - line 196: max unpooling transfers the argmax knowledge of maxpooling to the decoder. Do you use this behavior too? - Table 1: Should EX/NORB really be bold-faced? Is the diff between 0.59+/-0.12 really statistically significant from 0.58+/-0.11? - line 213: are all feature spaces well-suited for 1-NN? If a feature space is not close to a spherical Gaussian, it may perform poorly. If feature dimensions are individually standardized, it would avoid this issue. - It was a bit unclear how canonical samples were constructed on the face dataset ("least yaw displacement from a frontal pose"). This seems to require a lot of priors on faces and does not seem like purely unsupervised learning. Did the other competing methods require canonical examples to be designated?

Reviewer 3



The authors propose a manner of weak self-supervised learning, with a novel loss function, which allows the learning of an embedding that is robust to transformations of the inputs. In turn, this embedding facilitates downstream (classification) tasks, allowing for a low sample complexity. The paper presents a few very interesting findings. For one, the proposed embedding, learned with the joint loss (OJ), performs better at the even/odd MNIST transfer learning task. Additionally, and perhaps most importantly, the authors show a trade-off on the Late Night faces dataset between the number of orbits used to learn an embedding, and the number of samples needed for learning. This finding connects very well to the intuition that more visual experience would better prepare a learner for any specific downstream tasks, hence requiring fewer samples for learning. Perhaps my main point of criticism is based on the same figure 4, but then on the left. The number of samples for learning is on the x-axis, the classification accuracy on the y-axis. If we select a reasonable classification accuracy (e.g., over 90%?), the curves of the different learning losses come very close together. Notably, having the joint loss (OJ) instead of its components does not seem to matter at such a point anymore. It thus raises the question: does the loss then actually matter so much? A related remark can be made about the trade-off graph: the interesting part may be the yellow part. The x-axis of the right figure stops before we can see that increasing the number of orbits actually would give us such a performance. It would be good if the authors clarified this matter. Furthermore, the statistical tests in Table 1 are hard to believe. The caption mentions that bold signifies a significant difference from OJ. However, how can ST on MINST with a performance of 0.97 +- 0.02 be not statistically different from OJ with 0.67 +- 0.05? Perhaps ST was not included in the tests? Still, a similar strange result then is the comparison with OE in the bottom 3 rows. Especially the result of OE on the M-PIE set with 0.61 +- 0.01 as compared to OJ’s 0.95 +- 0.01. I would urge the authors to reconsider their statistical test, or convincingly show how distributions with such a different mean and small standard deviation can still be regarded as the same… In general, I think this is an original and well-written paper, so I would recommend an accept. Small remarks: 1. Definition 3: should you not also add = x to the formula xc = g0 x [ = x], given that g0 is supposed to be the identity transformation? 2. “Note that the loss is independent of the parameterization of Phi, Phi~”. What do you mean with parameterization, exactly? The loss function clearly depends on the parameter instantiation of Phi, Phi~. The way in which Phi is parameterized? But why then say that it does depend on the actual selection of triplets? 3. Proof of proposition 1. Frankly, I do not really see why this proposition is of real interest… It seems rather circular. Also, is the condition of boundedness easily (provably) satisfied, and is it then still meaningful? Why does the sqrt(alpha) go away? Is the ||…||2 the square, and not the 2-norm? 4. I think the remark on a generalization of de-noising auto-encoders is interesting. 5. Exemplar (EX): is it possible in one phrase to give an intuition how this method works? 6. “consisted from” -> “consisted of”